# Visible light reduces *C. elegans* longevity

C. Daniel De Magalhaes Filho[1,2], Brian Henriquez[2], Nicole E. Seah[3], Ronald M. Evans[2], Louis R. Lapierre[3] & Andrew Dillin[1]

The transparent nematode *Caenorhabditis elegans* can sense UV and blue-violet light to alter behavior. Because high-dose UV and blue-violet light are not a common feature outside of the laboratory setting, we asked what role, if any, could low-intensity visible light play in *C. elegans* physiology and longevity. Here, we show that *C. elegans* lifespan is inversely correlated to the time worms were exposed to visible light. While circadian control, *lite-1* and *tax-2* do not contribute to the lifespan reduction, we demonstrate that visible light creates photooxidative stress along with a general unfolded-protein response that decreases the lifespan. Finally, we find that long-lived mutants are more resistant to light stress, as well as wild-type worms supplemented pharmacologically with antioxidants. This study reveals that transparent nematodes are sensitive to visible light radiation and highlights the need to standardize methods for controlling the unrecognized biased effect of light during lifespan studies in laboratory conditions.

[1] The Howard Hughes Medical Institute, Molecular and Cell Biology Department, Li Ka Shing Center, University of California Berkeley, Berkeley, CA 94720, USA. [2] The Salk Institute for Biological Studies, Gene expression laboratory, The Howard Hughes Medical Institute, 10010 N.Torrey Pines Road, La Jolla, CA 92037, USA. [3] Department of Molecular Biology, Cell Biology and Biochemistry, Brown University, Providence, RI 02912, USA. Correspondence and requests for materials should be addressed to A.D. (email: dillin@berkeley.edu)

From plants to mammals, the ability to sense light is a ubiquitous feature of organisms that serves a broad range of functions including energy synthesis, DNA repair, modulation of circadian rhythms, and informing on the environment. Historically, the nematode *Caenorhabditis elegans* was believed to lack the ability to sense light due to the absence of a *bona fide* photoreceptor system and its original isolation in soil samples. However, recent work in *C. elegans* has identified the LITE-1 taste receptor homolog as a UV-specific photoreceptor totally distinct from other photoreceptors found in metazoans, microbes, and plants[1]. Interestingly, high-energy UV and blue wavelength light trigger an escape behavior and a pharyngeal pumping (feeding) inhibition in *C. elegans*[2–6]. Notably, both behaviors are optimized for intense ultraviolet/purple light, but not wavelengths of the visible spectrum. While the ecological role of light on pumping inhibition remains unclear, the negative phototaxis would have been selected during evolution to maintain worms in dark places, either in the soil during the day or outside during night time, thereby protecting them from UV-mediated cellular damage[4–6]. Indeed, in contrast to animals with external pigmentation, the transparent body of nematodes allows light to penetrate their body, making them particularly vulnerable to the mutagenic effects of UV[7, 8]. At the cellular level, UV light is mainly perceived by the ASJ, ASK, and AWB ciliary sensory neurons through mechanisms that are not completely understood. In the ASJ

neurons, LITE-1 transduces the light signal via G-protein signaling, resulting in a downstream signaling cascade involving the TAX-2 cGMP-sensitive CNG channels[5, 6, 9].

The identification of light-responsive genes in *C. elegans* has led to the hypothesis that, similar to other organisms, light perception could entrain circadian rhythms by acting as a zeitgeber (time giver). Circadian rhythms are endogenous rhythms of approximately 24 h that help organisms synchronize their physiology and behaviors with the daily alteration of light and dark phases from the Earth's rotation. Such rhythms still persist in constant darkness (free-running conditions), and can be reset by exposure to external signals including light (entrainment). In humans, disruption of circadian rhythms through light exposure at night, as experienced by shift workers or chronic jet lag, has been linked to detrimental impacts on eating time, obesity, diabetes, and cancer incidence[10–12]. Moreover, disruption of the circadian cycle has been shown to have a profound impact upon the lifespan of Drosophila[13–15]. Recent reports indicate that *C. elegans* entrained to 12 h/12 h light/dark cycles possess circadian rhythms upon gene expression, locomotor activity, and resistance to osmotic and oxidative stress[16–21]. Interestingly, light also drives the expression of a large group of genes that are not true circadian genes since they do not continue to cycle in dark/dark, free-running conditions[21], which suggests that light could also

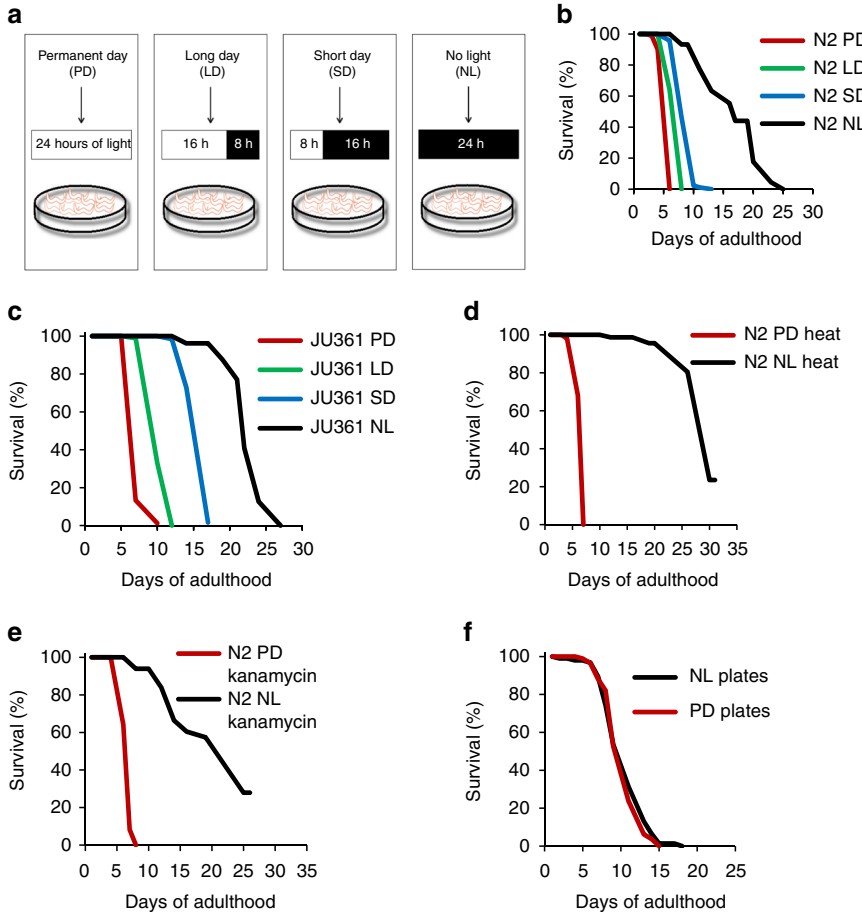

**Fig. 1** Daily light exposure decreases *C. elegans* lifespan. **a** Experimental setup for exposing worms to different photoperiods with the amount of light indicated in the white box and the amount of darkness indicated in the black box for each condition. **b** Lifespan of N2 worms and (**c**) of the JU361 wild isolate under the four different photoperiods. **d** Lifespan of N2 worms under PD or NL conditions with heat-inactivated bacteria as the source of food or, (**e**), antibiotic-lyzed bacteria as the source of food. **f** Lifespan of N2 worms kept in NL conditions after having been placed from D1 adult on plates previously exposed to PD or NL conditions. Worms were transferred every 1–2 days until the end of their lifespan

| | Permanent-day incubator | Long-day incubator | Short-day incubator | Constant dark[a] | ANOVA |
|---|---|---|---|---|---|
| **Table 1 Temperature of plates in incubators set at 22 °C and under different photoperiods** | | | | | |
| Surface of agar-based medium | $23.0 \pm 0.3$ | $23.2 \pm 0.2$ | $23.4 \pm 0.2$ | $23.4 \pm 0.2$ | $P = 0.507$ |
| Within the agar-based medium | $23.1 \pm 0.2$ | $22.8 \pm 0.2$ | $22.9 \pm 0.2$ | $23.1 \pm 0.2$ | $P = 0.828$ |

[a]Plates were kept in a black box covered with aluminum foil in the permanent-day incubator

have circadian rhythm-independent effects. Thus, the effects of light on worm physiology are just beginning to be clarified.

We used *C. elegans* to test whether the photoperiod (the interval in a 24-h period during which an animal is exposed to light) could impact its physiology and lifespan. This is also of special interest since standard laboratory manuals and practices for *C. elegans* handling completely ignore random exposure to light (laboratory environment, dissecting microscopes) versus dark (incubators). Here, we demonstrate that daily exposure to white light decreases *C. elegans* lifespan and alters development. Importantly, these effects are not mediated through known photoreceptor pathways or through a proper disruption of circadian rhythms. Our results indicate that the effect of light on *C. elegans* lifespan is not specific to a particular wavelength of the visible spectrum, but is photon energy dependent. We find that light exposure causes oxidative stress and induces canonical stress responses. Several long-lived mutants that ectopically activate these stress-responsive pathways are resistant to light stress. Furthermore, we find that treatment of wild-type worms with antioxidants is sufficient to rescue their short lifespan due to light exposure.

## Results

**Daily light exposure decreases *C. elegans* lifespan**. The natural habitat of *C. elegans* is unclear with suggestions for it living solely in the soil, while others suggest a terranean environment[22]. Although no *bona fide* photoreceptor has been identified to date, previous work has shown that light can entrain a circadian rhythm in *C. elegans*, similar to several other organisms including mammals[21]. While light exposure could entrain a set of genes that appear circadian in nature, a large group of genes regulated by light are allegedly noncircadian[21]. To investigate whether different photoperiods could affect *C. elegans* lifespan, we compared the survival of wild-type N2 worms placed during their entire life in the following conditions: 1) constant darkness (NL) with 0 h of light/24 h of darkness each day, or 2) permanent-day light (PD) regimen with 24 h of light/0 h of darkness per day (Fig. 1a) using fluorescent white light. In all experiments, temperature was controlled to 22°C at all times. Strikingly, worm lifespan was dramatically reduced under PD conditions with a mean lifespan of $4.8 \pm 0.1$ days (Fig. 1b), while under NL, the mean lifespan of N2 worms was $14.7 \pm 0.5$ days.

Intrigued by the robust lifespan difference due to constant light exposure (PD), we reasoned that permanent exposure to light is an artificial situation that worms do not encounter in the wild, and that exposing worms to photoperiods closer to the ones found in England, where N2 worms were initially isolated, would more closely mimic their natural lifespan. We introduced a short-day photoperiod (SD) consisting of 8 h of light/16 h of darkness per day, thus reproducing the shortest days of the year at the latitude of Cambridge, England, and a long-day photoperiod with 16 h of light/8 h of darkness similar to the longest days of the year at the same latitude. We found that SD and LD photoperiods resulted in intermediate lifespan between NL and PD conditions (Fig. 1b). Worms under SD condition lived on average for $7.9 \pm 0.1$ days, while worms under LD condition had a mean lifespan of

$6.3 \pm 0.1$ days (Fig. 1b). Thus, it appears that *C. elegans* lifespan is inversely proportional to the photoperiod in which they reside.

Because N2 worms were isolated in 1951 and several decades of passage in the laboratory might have fixed some traits, including lifespan, with adaptation to the permanent dark condition of the laboratory incubators where worms are maintained most of their life[23], we sought to test a recently isolated wild strain of *C. elegans*. Similar to N2 worms, we found that the JU361 wild-isolate strain (which was isolated in France in 2002) had a lifespan inversely proportional to its photoperiod exposure (Fig. 1c).

There are several indirect reasons to explain the proportional decrease in the lifespan of worms exposed to white light. One, exposure to light could heat the plates, agar, bacteria (food source), and worms, thus shortening their lifespan. Using a precision thermometer, we tested whether the SD, LD, and PD photoperiod could increase the temperature of the agar plates, but found no difference (Table 1). Two, different photoperiods could indirectly affect *C. elegans* lifespan by acting primarily on the bacteria that serve as the food source for the worms. Hence, while circadian rhythms *per se* have not been reported for *E. coli*, it is possible that light impaired the bacterial physiology, making themselves a detrimental food source for the worms. To test this possibility, we measured the lifespan of N2 worms exposed to PD in which the bacteria were killed by either antibiotics or heat. As shown in Fig. 1d, e, *C. elegans* lifespan was still dramatically reduced under the permanent light photoperiod when fed with dead bacteria, with an average lifespan of 5.7 days for both heat-killed bacteria and kanamycin-killed bacteria, almost identical to conditions with live bacteria. We extended these findings to include a possible photochemical damage reaction on the agar-based media on which bacteria grow and *C. elegans* is cultured upon. We exposed bacteria-seeded plates without worms to PD or NL conditions for 24 h. Thereafter, we transferred adult worms raised in the dark to the PD light pretreated plates and returned all plates to the dark. We repeated this manipulation every day or every other day, until the end of the animal's life. As shown in Fig. 1f, worms placed on plates previously exposed to PD or NL do not differ in lifespan. Therefore, the brevity of life span due to light exposure could not be caused by increased temperature or photoactivation of a lifespan-shortening factor in either the bacterial food or the agar plates. Taken together, these results indicate that daily light exposure directly decreases *C. elegans* lifespan proportional to the amount of light received per day.

**Light exposure affects development**. To better understand the potential cause of the short lifespan of worms exposed to visible light, we evaluated how light exposure could affect development. We found during the lifespan experiments described above that worms exposed to light from birth (PD) developed into slightly smaller adults compared to animals kept in constant darkness (Fig. 2a). Long-day and permanent-day conditions decreased the size of the animals by 13% and 17%, respectively, compared with no-light (NL) control animals (Fig. 2a). We found no significant difference between worms under SD and NL conditions. Therefore, permanent light exposure induces a developmental defect upon *C. elegans* growth.

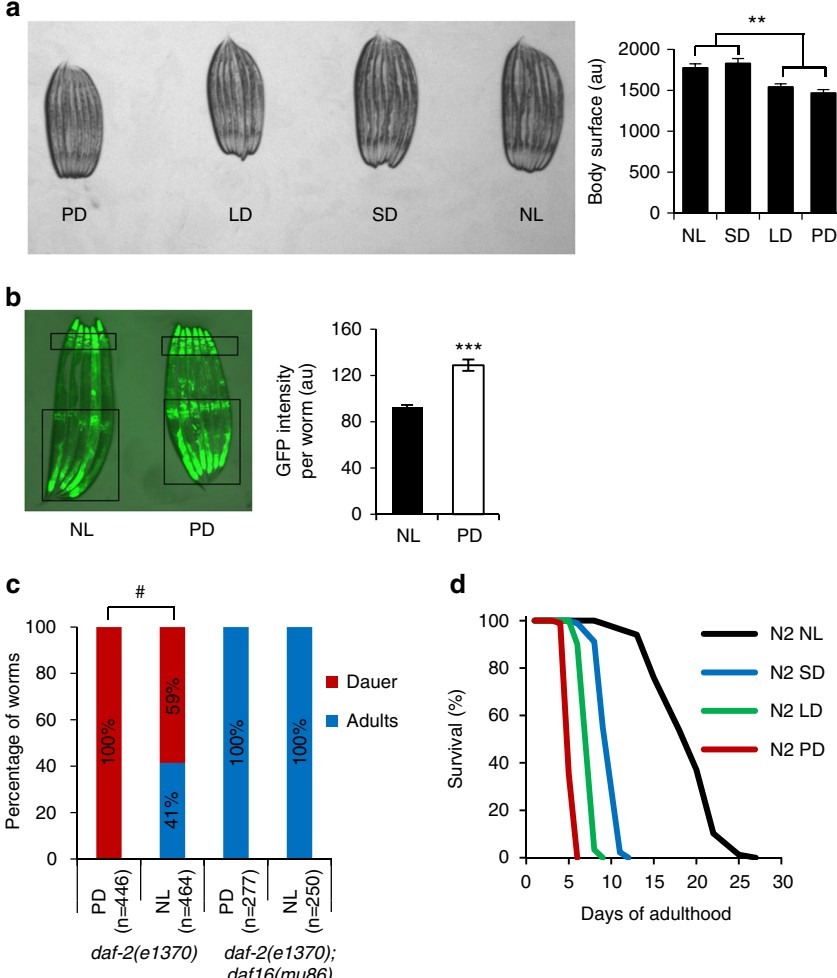

**Fig. 2** Light exposure affects *C. elegans* development. **a** Left, appearance of D1 adult N2 worms having been exposed to the four photoperiods from hatch; right, quantification of body surface from the left panel. **b** *sod-3p*::GFP worms exposed to permanent light or no light from hatch and observed at D1 adult under a GFP fluorescent filter (left) and quantification of the fluorescence (right); black rectangles indicate known areas of *sod-3p*::GFP expression where fluorescence was quantified. **c** *daf-2(e1370)* worms raised from hatch at 22 °C under permanent light or no light and quantified at +72 h for dauer or adult stage. *daf-2(e1370);daf-16(mu86)* strain was used as a negative control. **d** Lifespan of N2 worms raised until D1 adult under NL and from D1 onward under the four different photoperiods. In panels a and b, bars represent mean ± SEM, \*\*P < 0.01; \*\*\*P < 0.001 using Student's *t* test. #, P < 0.001 using Fisher's exact test

The activation of the forkhead transcription factor DAF-16, which translocates from the cytoplasm to the nucleus upon a broad range of environmental stresses, regulates stress responses, development, and growth[24]. Using a *sod-3p*::GFP reporter strain as a readout of DAF-16 activation[25], we observed robust induction of the *sod-3p*::GFP reporter in worms exposed to light from hatch as opposed to animals developed in the dark (Fig. 2b). Endogenous *sod-3* mRNA levels were also increased upon PD treatment (see below). As an additional measure of DAF-16 activation due to light exposure, we followed the temperature-sensitive dauer-constitutive *daf-2(e1370)* mutants. Worms harboring this mutation develop normally to adulthood when cultured at the permissive temperature of 15 °C. However, at the nonpermissive temperature of 25 °C, loss of *daf-2* function results in reduced insulin/IGF-1-like signaling, robust DAF-16 hyperactivation, and transition of larval worms into the dauer larval stage[26, 27]. The *daf-2(e1370)* mutant allele is a hypomorphic allele that is sensitive to additional environmental stressors as well as genetic modifiers. At 22 °C, DAF-2 signaling is only partially decreased by the *daf-2(e1370)* mutant allele, and in constant darkness, only 41% of *daf-2(e1370)* mutant worms entered the

dauer diapause stage (Fig. 2c). However, at the same temperature, but under PD condition, 100% of the *daf-2(e1370)* mutant worms entered the dauer stage (Fig. 2c). The increased dauer entry caused by PD upon the *daf-2(e1370)* mutation was fully dependent upon *daf-16* as none of the *daf-2(e1370);daf-16(mu86)* double-mutant animals exposed to PD entered the dauer stage (Fig. 2c). Again, increased temperature due to PD condition was tested and ruled out (Table 1). Collectively, these results indicate that light exposure from hatching creates a developmental stress to worms, which in part, activates DAF-16.

**The lifespan-shortening effect of light exposure is not confined to development.** Exposure to white light induced growth delay and activation of DAF-16, suggesting that the lifespan-shortening effects caused by white light exposure could be due to pleiotropic defects in development caused by light exposure. To test this hypothesis, we treated animals with white light post development and scored their lifespan. We raised animals in the dark before exposing them from the first day of adulthood to the four different photoperiods: NL, SD, LD, and PD, until the end of their

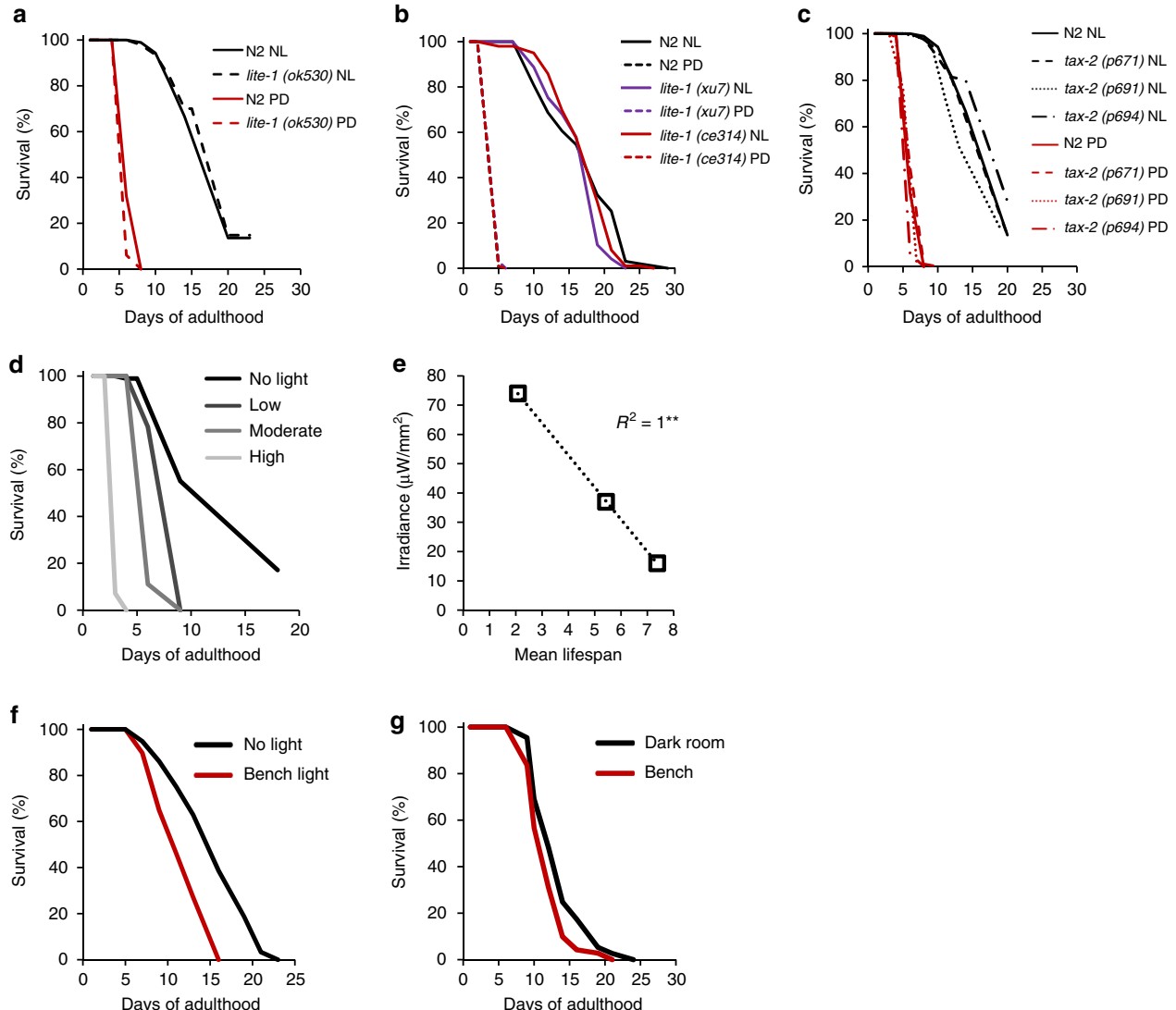

**Fig. 3** Light intensity is the primary signal decreasing the lifespan. **a** Lifespan of *lite-1(ok530)* mutants and **b** *lite-1(xu7)* and *lite-1(ce314)* under PD or NL conditions. **c** Lifespan of *tax-2* mutants is reduced upon daily light exposure. **d** Lifespan of N2 worms is proportional to the intensity of white light received (permanent-day condition). **e** Relationship between light intensity and mean lifespan; the black dotted line represents a linear trendline of the series. **f** Lifespan of worms exposed to laboratory light versus worms kept in the dark. **g** Lifespan of worms scored and transferred in the dark under dim stereomicroscope light, versus worms scored and transferred under laboratory light. \*\*$P < 0.01$ using Pearson's coefficient

life. As shown in Fig. 2d, worms raised from early adulthood under NL live significantly longer than their counterparts exposed to SD, LD, and PD conditions from early adulthood. Moreover, the lifespan-shortening effects of light exposure confined to adulthood were very similar to the effects observed in animals exposed during both development and adulthood (Figs. 1b, c and 2d). Therefore, the life-shortening effect of light exposure can be uncoupled from its effect on development.

**Light intensity is the primary signal decreasing lifespan.** *C. elegans* senses UV light through the UV photoreceptor LITE-1 and the downstream signaling protein TAX-2, which, at very high-light intensities trigger an escape behavior[5, 6] and decreased pharyngeal pumping (eating)[3]. TAX-2 and LITE-1 have also been shown to transmit light information to the circadian clock since light fails to entrain rhythmic gene expression in mutant worms for these genes[16, 21]. We tested whether the lifespan-shortening effect of light exposure was due to activation of either *tax-2* or *lite-1*. We found that *lite-1(ok530)* mutant worms under PD

conditions lived shorter (−70% mean lifespan) than their counterparts in NL conditions (Fig. 3a). We also tested two other *lite-1* null mutant strains and similarly found that, even in the absence of the LITE-1 photoreceptor, light exposure strongly decreased their lifespan (Fig. 3b). The same was true when we utilized three different *tax-2* mutant strains (−63 to −72% mean lifespan, Fig. 3c). Thus, the effect of light on lifespan is not mediated through known photoperception pathways in the worm, nor appears to be related to a circadian rhythm effect.

Having excluded the potential involvement of known UV light sensory genes and circadian rhythm effects in the lifespan phenotype that we observed, it remained possible that different photoperiods decrease the lifespan via: (a) a light-driven but circadian rhythm-independent manner, and in that case, light would be sensed by a receptor different from LITE-1 and TAX-2; or (b) through a nonphotoreceptor effect with light acting physically as a toxic stress to cell components. The later hypothesis predicts that at constant photoperiod, the brighter

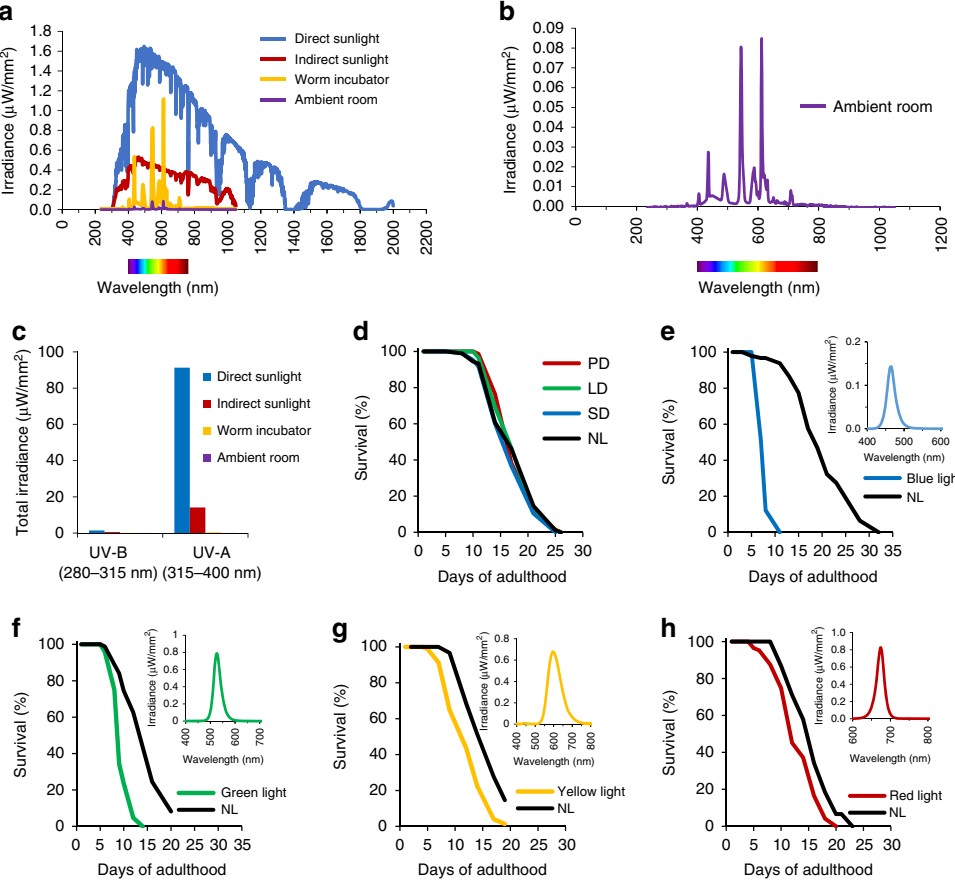

**Fig. 4** Worm lifespan is inversely proportional to wavelength energy. **a** Irradiance spectrum of the sun under direct or indirect sunlight, compared to the spectrum of white fluorescent tubes used in incubators and the laboratory (ambient room). **b** Detailed irradiance spectrum of white fluorescent tubes used in the laboratory (ambient room). **c** Total irradiance of UV light emitted by fluorescent white light versus solar radiation. **d** Lifespan under dark conditions of progeny derived from adult worms exposed to different photoperiods. **e** Lifespan of worms exposed permanently to blue, (**f**) green, (**g**) yellow, or (**h**) red LED light compared to controls kept in NL condition. Insets represent the spectrum of each color-specific LED

the intensity of light, the shorter the lifespan; and at constant intensity, the longer the photoperiod, the shorter the lifespan.

We previously showed in Fig. 1 that at constant white light intensity, worms exposed to PD lived shorter than worms on LD, which lived shorter than those on SD, which themselves lived shorter than worms kept in NL condition. To further test whether the duration of light exposure had a quantal effect on lifespan, we compared the lifespan of adult worms exposed to four different intensities of white light under PD conditions: high-light intensity (HI), moderate-light intensity (MI), low-light intensity (LI), and zero light (NL condition). The results shown in Fig. 3d confirm that worm lifespan is inversely proportional to the brightness of the light that they are exposed to since worm lifespan follows HI<MI<LI<NL. Strikingly, the light intensity and the mean lifespan show a nearly perfect linear relationship (Fig. 3e, $R^2 =$ 1.000, $P < 0.01$ using Pearson's correlation coefficient). Unequivocally, it is the total dose of light irradiation per se that is reducing the lifespan in *C. elegans*.

To the best of our knowledge, general guidelines of *C. elegans* cultivation during lifespan experiments do not indicate that careful control of light exposure is required to ensure the reproducibility of lifespan experiments. Based on our observations that light decreases the lifespan in *C. elegans*, and especially even low-light intensity can shorten the lifespan, we investigated whether the low-intensity white light found in a general laboratory could affect the lifespan when compared to worms

kept constantly in the dark under the same temperature conditions. As shown in Figure 3f, worms kept constantly on the bench under lab light (7 a.m. to 9 p.m. daily) live significantly shorter (−26% mean lifespan, $P < 0.001$ using log-rank test) than worms kept on the same bench but covered with aluminum foil to prevent light exposure. Struck by the difference in lifespan of these worms cultured side by side, one receiving light exposure and the other not, we examined the minimal amount of light stress required to produce a significant effect on the lifespan. We compared the survival of worms kept in NL conditions but scored and transferred to new plates in a dark room, under a very dim stereomicroscope light to reduce light exposure (control group). In parallel, we performed the same experiment, but scored survival and passaged worms to new plates under laboratory light conditions using normal stereomicroscope light. In this experimental group, worms were exposed to lab light on an average of about 20 min per daily manipulation. Surprisingly, we found that worms exposed to lab light during scoring and passaging lived slightly, but significantly and reproducibly, shorter than worms scored and transferred in the dark room (Fig. 3g). Collectively, these findings indicate that light is a toxic stress to worm lifespan independent of circadian rhythm and dependent upon length of light exposure and intensity, suggesting that photooxidation is a major driver in this process.

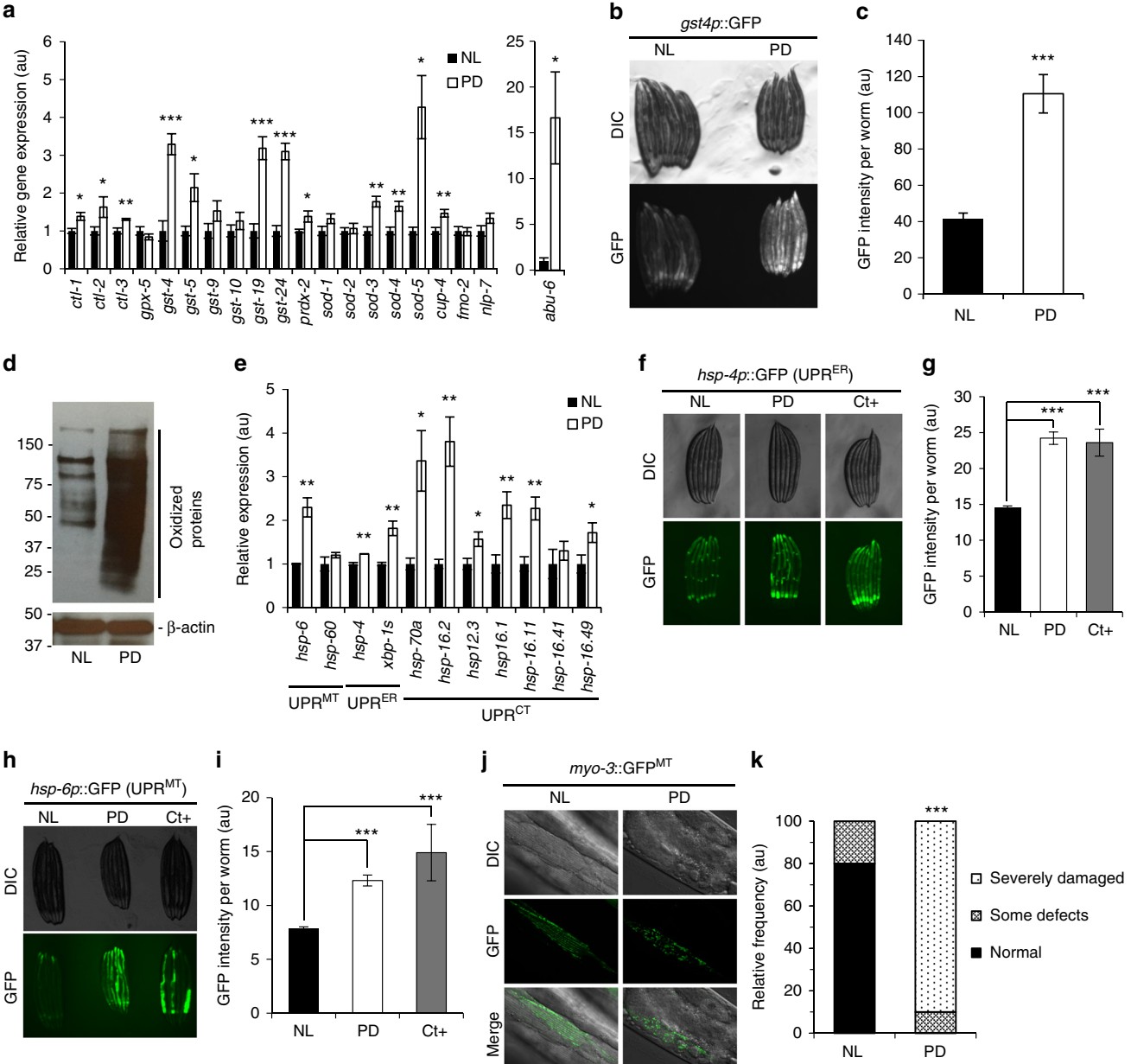

**Fig. 5** Visible light creates photooxidation in *C. elegans*. **a** Relative expression levels of enzymes involved in oxidative stress response. **b** Microscope pictures of *gst-4p*::GFP worms exposed to PD or NL condition and observed under DIC lens (top panel) or GFP fluorescence (bottom). **c** Quantification of the GFP intensity from the experiment in **b**. **d** Oxyblot from worms exposed during 4 days to PD or NL. The top panel reveals DNP-derived proteins, as an indicator of oxidized proteins; β-actin antibody was used to verify equal loading (bottom panel). **e** Relative expression levels of mitochondrial UPR (UPR^MT), endoplasmic reticulum UPR (UPR^ER), and cytosolic UPR (UPR^CT) genes between worms exposed to PD or NL condition from hatch to D1 adult. **f** Microscope pictures of *hsp-4p*::GFP reporter worms under NL and PD conditions. Tunicamycin was used as positive control (Ct+) for induction of the ER-UPR[40]. **g** Quantification of the GFP intensity from the experiment in **f**. **h** Microscope pictures of *hsp-6p*::GFP reporter worms under NL and PD conditions. *cco-1* RNAi was used as positive control (Ct+) for induction of the mito-UPR[39]. **i** Quantification of the GFP intensity from the experiment in **h**. **j** Microscope pictures of *myo-3*::GFP^MT worms after 7 days of PD or NL condition and observed under DIC or GFP fluorescence with 63X objective; and **k** quantification of the overall mitochondrial integrity from the experiment in panel **j**. Bars represent mean ± SEM, *P < 0.05; **P < 0.01; ***P < 0.001 using Student's *t* test except for **k**: Fisher's exact test

**Worm lifespan is photon energy dependent**. Light toxicity, or phototoxicity, is a phenomenon that has been well described for the nonvisible, shorter wavelengths of the light spectrum, especially ultraviolet (UV) and gamma radiation upon organismal physiology[28–33]. Hence, wavelengths shorter than 200 nm are considered as ionizing radiation because the energy carried by photons of this wavelength is sufficiently powerful to dislodge electrons from their orbital, creating an ionic molecule that is highly reactive. UV light defines wavelengths falling between 200 nm and 400 nm. Following UV photon absorption, DNA bases are excited which can result in pyrimidine dimers causing a DNA damage response and mutagenesis[28–30].

Under laboratory conditions, worms are exposed to a white fluorescent light. We measured the spectrum of this "white" light

to better understand the relative distribution and intensity of wavelengths. We also compared this spectrum to the solar radiation spectrum found at the surface of Earth in La Jolla, California under direct and indirect sunlight. As presented in Fig. 4a, b, c, white fluorescent light has no detectable ionizing radiation and barely detectable UV radiation, especially when compared to sunlight. To test if the small amount of UV radiation present in white fluorescent light could be involved in the short *C. elegans* lifespan due to random mutagenesis, we tested whether the short lifespan of animals exposed to fluorescent white light was heritable. In contrast to a role of mutagenesis causing the short lifespan, we found that the progeny from parents cultured in SD, LD, and PD photoperiods, had a normal lifespan when raised in dark conditions (Fig. 4d). Therefore, the nearly negligible UV radiation found in white light is not enough to explain the short lifespan of animals cultured under light conditions.

Since a large spectrum of wavelengths are present in white light, we tested whether certain wavelengths of the visible spectrum could explain the lifespan-shortening effects of white light exposure. Using color-specific LEDs to emit only a narrow range of the visible spectrum, we found that worms exposed during their entire life to blue light ($470 \pm 30$ nm) in permanent illumination conditions were short lived compared to worms kept in NL conditions (Fig. 4e). Their mean lifespan was reduced to $7.3 \pm 0.1$ days under blue light compared to $19.2 \pm 0.6$ days for controls kept in the dark. Under green light ($530 \pm 30$ nm) in permanent illumination conditions, worms were also short lived, however, the lifespan of these worms was slightly longer ($8.7 \pm 0.3$ days, Fig. 4f). Longer wavelengths such as yellow ($600 \pm 60$ nm) and red ($675 \pm 20$ nm) in permanent illumination were also sufficient to decrease the lifespan with the respective mean survival of $11.3 \pm 0.4$ days (Fig. 4g) and $13.2 \pm 0.4$ days (Fig. 4h). Collectively, these results indicate that the detrimental effect of white light upon lifespan is not specific to a particular wavelength within the visible spectrum, but rather to the quanta of energy found within these particular wavelengths. By exclusion, these results also reject the hypothesis that the minimal amount of UV radiation present in fluorescent white light caused the short lifespan of worms. Finally, since worm lifespan is decreased under four different types of wavelength, it strongly argues against the dependence of a single photoreceptor-dependent mechanism as already suggested above.

**Visible light creates oxidative stress and a multicompartment UPR.** Light radiation in the visible range, even if less powerful than UV radiation, can carry enough energy to drive a photochemical reaction. This phenomenon requires two steps: first, that the reacting molecule in the worm absorbs the light at a specific wavelength, and second, that the energy state of this activated molecule is sufficiently high to result in a chemical reaction with another molecule. Photochemical reactions in the presence of oxygen lead to the generation of reactive oxygen species (ROS) including singlet oxygen $^1O_2$, and then to oxidative reactions with molecules within the cell, a deleterious process known as oxidative stress[34–36]. We tested whether the decreased lifespan of worms exposed to white light could be due to an increase in oxidative stress.

We first measured canonical oxidative stress genes belonging to the catalase, glutathione, peroxiredoxin, and the superoxide dismutase families, all involved in cell detoxification processes of oxidatively produced compounds and whose expression increases under oxidative stress[37–40]. As shown in Fig. 5a, expression levels for genes belonging to these families were strongly and significantly elevated under PD condition versus NL.

Using *gst-4p*::GFP reporter worms, we confirmed that *gst-4*, a major oxidative stress-responsive gene, was induced under PD conditions (+64% in fluorescence levels, $P < 0.001$, Fig. 5b, c). The increase in *sod-3* gene expression is also consistent with the higher fluorescence observed in *sod-3p*::GFP worms (Fig. 2b). Moreover, among other genes previously reported to be induced under oxidative stress, *cup-4*[40], encoding a ligand-gated ion channel similar to the nicotinic acetylcholine receptors, and *abu-6*[39], a UPR-independent transmembrane protein linked to ER stress[41], were both strongly increased under PD conditions, while *fmo-2*[37] and *nlp-7*[40] expressions were unchanged (Fig. 5a). Altogether, these results strongly suggest that visible light creates oxidative stress in worms.

To directly assess ROS-mediated damages, we analyzed protein oxidation and found that worms exposed to PD displayed much greater quantities of carbonylated proteins (Fig. 5d). Finally, to further test whether light creates an oxidative stress in worms, we verified whether the unfolded-protein response (UPR) was induced. Under general cellular stress, including sustained oxidative stress[42], damage to proteins occurs and triggers the UPR in one or several cellular compartments[43]. At the transcriptional level, *hsp-6* involved in the mitochondrial UPR (UPR$^{MT}$)[43, 44], was upregulated in PD compared with NL condition (Fig. 5e). Light exposure also results in endoplasmic reticulum UPR (UPR$^{ER}$) induction, as revealed by elevated levels of *hsp-4* and spliced *xbp-1* transcripts, and the cytosolic UPR (UPR$^{CT}$) as revealed by increased levels of *hsp-70a* and *hsp-16.2* transcripts as well as other members of the HSP family (Fig. 5e). In accordance with the above results, the stress elicited by light exposure is very different from a heat stress since the increase in *hsp-70a* and *hsp16.2* expression under PD during 72 h, even if significant, was several magnitudes lower than the induction achieved after only 1 h at 37 °C (Supplementary Figure 1). Independently, we confirmed the induction of the UPRs using fluorescent reporter worm lines: *hsp4p*::GFP[45] (Fig. 5f, g), and *hsp-6p*::GFP (Fig. 5h, i) worms. The lifespan reduction in worms exposed to light infers that even in the presence of the stress response, the intensity and/or duration of the insult overcomes the defense capacity of the animal.

Mitochondria are critical organelles for various cellular functions including energy production. After 7 days of permanent light exposure, mitochondrial integrity was severely compromised as revealed by fluorescent micrographs of muscle mitochondria using *myo3p*::GFP$^{MT}$ reporter worms (Fig. 5j, k)[46, 47]. Taken together, these results indicate that light exposure creates photooxidative reactions in worms, resulting in expression of antioxidant defenses and UPRs in different compartments of the cell, and ultimately structural damage to the cell, including mitochondria.

**Resistance to oxidative stress protects from light-deleterious effects.** To further understand how ROS, along with protein-folding defects, contribute to the light-mediated shortening of the lifespan, we aimed at rescuing the short-lived phenotype using several approaches. First, we tested whether two separate longevity interventions: decreased insulin/IGF-1-like signaling and partial loss of function in the mitochondrial ETC, both known to increase the lifespan and confer resistance to oxidative stress[38, 48–50], could protect worms from light stress. Importantly, we found that *daf-2(e1370)* mutant worms were protected from the light insult. Under SD condition, *daf-2* worms lived 87% longer ($P < 0.0001$) than N2 worms exposed to the same photoperiod (Fig. 6a, Supplementary Figure 2a and Supplementary Table 1). This increase was still present (+88%, $P < 0.0001$) when both strains were studied under LD conditions. Finally, under the

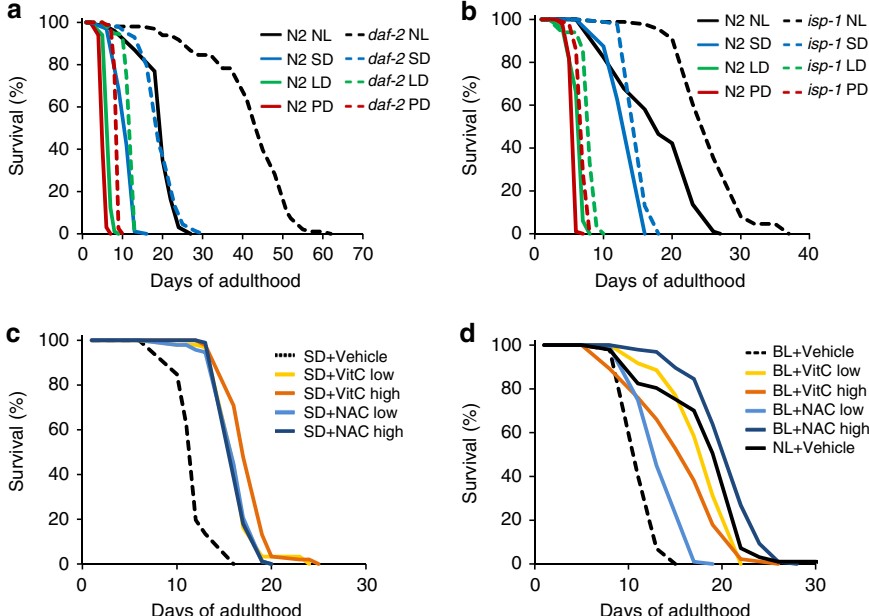

**Fig. 6** Increasing defense mechanisms protects *C. elegans* from light toxicity. **a** Lifespan of *daf-2*(e1370) mutants versus N2 worms under PD, LD, SD, and NL photoperiods. **b** Lifespan of *isp-1*(qm150) mutants versus N2 worms under PD, LD, SD, and NL photoperiods. **c, d** Lifespan of N2 worms supplemented with low (5 mM) or high (10 mM) concentration of VitC or NAC versus vehicle under SD condition (**c**) or bench light (BL) condition (**d**); NL+ vehicle was used as additional control

severe conditions of PD, *daf-2*(e1370) mutant worms lived longer than N2 worms, with a lifespan extension of +3.4 days (+77%, P < 0.0001) (Fig. 6a and Supplementary Table 1). We note that the resistance of *daf-2* worms to light damage is only partial since their lifespan under light exposure is shorter when compared to *daf-2* worms kept in the dark, indicating that *daf-2* mutant animals do not possess the full repertoire of protective mechanisms to combat light exposure.

We also investigated the lifespan of *isp-1*(qm150) mutants, which harbor a mutation in complex III of the mitochondrial ETC, making them protected from oxidative stress and long lived[51]. The lifespan of these animals was longer than the lifespan of N2 worms under SD, LD, PD, and NL conditions (Fig. 6b, Supplementary Figure 2b), with a respective increase of +14%, +25%, +27%, and +44% versus N2 worms (P < 0.0001 for all conditions).

Finally, we hypothesized that if the increased resistance to oxidative stress was the main explanation for conferring the mutant worms described above a longer lifespan under the different light paradigms, then, supplementing wild-type worms with antioxidants should similarly rescue their lifespan. Therefore, we treated N2 worms from hatch with N-acetyl cysteine (NAC) or vitamin C (VitC), each at two concentrations previously shown to confer resistance to oxidative stress[44, 52–55]. Under SD light condition, NAC and VitC of both low and high concentrations strongly increased the lifespan when compared to vehicle-treated worms (Fig. 6c). The lifespan rescue with antioxidants was even more spectacular when exposing worms to a weaker white light intensity. To this end, we used the bench light condition, previously described to reduce the lifespan of worms by 25% (Fig. 3f). Worms exposed to indoor fluorescent light while on plates supplemented with NAC or VitC at high or low concentrations all demonstrate a significantly longer lifespan than worms not receiving antioxidants (+30 to +105% increase in mean lifespan, Fig. 6d). Strikingly, when comparing the lifespan of NL control worms to the lifespan curves of worms under bench

light with antioxidants, we note that VitC at low concentration is rescuing almost entirely the lifespan of bench light worms and that the lifespan rescue is complete using NAC at high concentration (Fig. 6d). Collectively, these data indicate that the damaging effects of photooxidation created by visible white light can be rescued by genetic or pharmacological manipulations increasing antioxidant defenses.

## Discussion

By aiming at studying how the modulation of *C. elegans* daily light exposure could impact the lifespan, we have discovered that photoperiod length negatively affects the lifespan. However, this effect is independent of circadian rhythm *per se* since we find that modulating the intensity of light while keeping the same total amount of daily light exposure is sufficient to modulate the lifespan. It is also independent from signaling through previously described photoreceptor cascades, such as *lite-1* and *tax-2*, since worm mutants for these genes are also short lived when exposed to light. Importantly, we find that the detrimental effect of visible light on *C. elegans* is directly dependent on the photon energy carried by light and the duration of exposure. By examining the larval development of worms exposed to light, we discovered that illumination also negatively affected growth, but this effect could be uncoupled from the lifespan-shortening effect of light since adult worms exposed to light retained a strong decrease in lifespan. We further gained insight into the mechanism of light-induced toxicity by finding that light irradiation results in oxidative stress to which worms respond by activating canonical stress response pathways. Accordingly, long-lived mutant worms with constitutive upregulation of these stress response pathways are protected from the light toxicity, albeit only partially. Remarkably, pretreatment of worms with antioxidants is sufficient to rescue their normal lifespan upon light exposure, with a stronger rescue effect observed under lower light intensities.

A major finding presented here is the discovery that visible light radiation at intensities comparable or below natural sunlight has dramatic effects on *C. elegans* physiology and lifespan. To our knowledge, this is the first report of an effect of white light exposure on worm lifespan or development. In previous studies, light intensity in the blue, purple, and green wavelength ($26–76 \times 10^{-3}$ mW/mm$^2$) was found to have effects on phototaxis, eating rates, and acute death. However, when wavelengths were applied at intensities comparable to sunlight, no phototaxis was observed and long-term effects on physiology were not investigated. Here, we find that mimicking the natural visible light using industrial fluorescent white light at 40 µW/mm$^2$ dramatically impacted worm lifespan. Interestingly, exposure of worms to the artificial white light found in a typical lab environment (2.5 µW/mm$^2$) several hours a day was sufficient to impact the lifespan. Such findings strongly invite a reconsideration of the standard methods of *C. elegans* handling, especially in the context of aging research and stress biology.

We find that light radiation confined to the visible range (400–700 nm), i.e., in the absence of UV, causes oxidative stress as indicated by the upregulation of oxidative stress genes and the high oxidation of proteins. While high-intensity UV-C light damages nucleic acids and creates pyrimidine dimer mutations, white light appears to spare nucleic acid integrity, and ensuing mutations, should they happen, do not reach a sufficiently high threshold to cause a heritable lifespan phenotype. Thus, the harmful effects of visible light appear to be sharply different from the ones triggered by UV-C light as suggested before[56]. The observation that several wavelengths of the visible spectrum, especially very distant (blue light = 470 nm, red light = 675 nm) can decrease worm lifespan is informative and suggests that the dramatic effects of white light on worm physiology and lifespan are likely the sum of individual effects from specific wavelengths of the visible spectrum reacting upon specific molecules. In the future, it will be informative to know which classes of molecules are damaged and affected by particular wavelengths of light in the visible spectra.

In this paper, we demonstrate that increasing the antioxidant defenses by pharmacological supplementation of vitamin C or *N*-acetyl cysteine, two well-characterized antioxidants, prevents the oxidative effects of light exposure in worms. Not surprisingly, this effect is better observed at lower light intensities, when the oxidative stress is milder, with worms supplemented with NAC or VitC under bench light condition getting a stronger lifespan rescue than worms supplemented with the same amount of antioxidants but exposed to SD condition. Therefore, mutant worms equipped with increased antioxidant defenses are likely to be resistant to the damaging effect of light. We argue that this oxidative stress resistance is a likely explanation for the observed increased lifespan of *daf-2* and *isp-1* mutants under light but we cannot exclude that other mechanisms such as increased UPR responses might be playing a role too. In the future, it would be therefore interesting to test whether the sole upregulation of a UPR (cytosolic, ER, or mitochondrial) would be sufficient to confer worms resistance to the damaging effect of light.

Overall, the data presented here reveal that visible light is sufficient to create oxidative stress in *C. elegans* and shorten its lifespan. This effect is not confined to a specific wavelength of the visible spectrum but is stronger for shorter, higher-energy wavelengths. Our study strongly encourages a reconsideration of the general *C. elegans* practices in laboratory environment to now include light exposure as an environmental signal that should be carefully controlled and reported as it represents a potential bias during stress and lifespan experiments.

## Materials and methods

**Strains.** Nematodes were maintained using standard methods[57]. Briefly, NG plates were prepared by pouring sterilely 10 mL of nematode growth media[57] into a sterile vented Petri Dish of $60 \times 15$ mm (E&K Scientific, Santa Clara, CA, USA). Plates were kept at 4 °C until use. Two days before use, NG plates were brought to room temperature and 50 µL of liquid culture of *E. coli* OP50 bacteria was poured sterilely into the center of the plates. OP50 lawn from seeded plates was allowed to grow for 2 days at room temperature. Then, worms were placed on the plates using a sterile platinum pick. Strains were obtained either from the Dillin lab or from the Caenorhabditis Genetics Center which is funded by NIH Office of Research Infrastructure Programs (P40 OD010440). A strain list is available in Supplementary Table 2.

**Light experiments.** Seeded NG plates containing worms were placed in Percival incubators (model AR66-L, Percival Scientific, Perry, IA, USA), maintained at 22 °C, and equipped with Phillips Alto F17T8/TL741 fluorescent white light (Philips Lightning Company, Somerset, NJ, USA). For permanent-day (PD) conditions, lights remained on continuously. For long-day (LD) conditions, lights were on between 6 a.m. and 10 p.m. and off between 10pm and 6am, resulting in a 16-h light/8-h dark photoperiod. For short-day (SD) conditions, lights were on between 10 a.m. and 6 p.m., and off between 6pm and 10am, resulting in an 8-h light/16-h dark photoperiod. For constant darkness (NL) conditions, plates were placed in a single layer in a black cardboard box covered with aluminum foil. This box was placed in the PD incubator at the same distance from the light source as the light-exposed plates.

For wavelength-specific experiments (blue, green, yellow, or red light), we used LED-equipped devices to restrict light emission to a narrow-wavelength range. For red and blue colors, we used the Quantum device controller QB-2200 along with the LED-light device QB-1310CS (QuantumDevices, Barneveld, WI, USA) to emit at $470 \pm 30$ nm (95% of total irradiance, blue light) or $675 \pm 20$ nm (90% total irradiance, red light) at user-defined intensities. For green and yellow lights, an in-house Salk Electronics Department built two separate devices with fans, a heat sink, and LEDs mounted on it, emitting either at $530 \pm 30$ nm (90% total irradiance, green light) or $600 \pm 60$ nm (90% total irradiance, yellow light). Each device was connected to a potentiometer controlling the intensity of the light emitted.

Irradiance, expressed as µW/mm$^2$/nm was recorded using a Luzchem spectroradiometer (Luzchem Research, Ontario, Canada). The spectrum and intensity applied in each light experiment are shown in Fig. 3. Irradiances (integrated between 400 and 700 nm) used for Fig. 2c are as follows: very low intensity = 2.5 µW/mm$^2$, low intensity = 16 µW/mm$^2$, moderate intensity = 37 µW/mm$^2$, and high intensity = 74 µW/mm$^2$. Solar reference spectrum G173-03 at sea level and measured on an inclined plane at 37° tilt toward the equator, facing the sun, was described by the American Society for Testing and Materials (http://rredc.nrel.gov/solar/spectra/am1.5/). The spectrum for indirect sunlight was obtained by measuring the irradiance in the shade at sea level in San Diego (CA, USA) on the 1$^{st}$ of April 2010 at noon.

**Lifespan analysis.** Synchronized animals were prepared by the egg-laying method by placing young adults for 4 h onto bacteria-seeded plates and subsequently removing them. All lifespan analyses were performed starting with 100 worms per condition and carried out at 22 °C unless otherwise specified. Viability was scored every 1–3 days, as previously described[58]. In brief, death was determined when worms did not respond to a gentle touch with a sterilized platinum wire. Worms were censored when missing, having crawled off, having burrowed, or if they displayed internal hatching or vulval rupture. Moreover, except for the experiment described in Fig. 2f, worms were scored under a regular stereomicroscope on the bench under general lab light environment, usually at around midday which corresponds to the light phase of each light treatment (SD, LD, and PD). To minimize time-out of the incubator, the scoring of worms was performed within 20 min per condition. Worms belonging to the NL group were kept in a dark box when removed from the incubator and brought to the bench for scoring. To minimize light exposure for these NL worms, plates were removed one at a time from the dark box and scored (which takes usually 1–2 min) before being immediately returned to the dark box. For the experiment described in Fig. 2f, worms were kept in a common worm incubator without light. On the day of scoring, they were either brought on the bench under general laboratory light to be scored under a regular stereomicroscope, or brought into a dark room to be scored under a regular stereomicroscope with its light set to the minimal intensity allowing the experimenter to observe worms and score them. We also estimate for this experiment that worms scored on the bench were exposed to general lab light for an average of 20 min per day of scoring, which is likely to be minimal compared to the amount of time that worms are typically exposed to lab light in *C. elegans* laboratories performing lifespan analyses. Worms scored in the dark room were exposed to the dissecting microscope light for an average of 1–2 min per day of scoring.

**Lifespan on dead bacteria.** To perform lifespan experiments on dead bacteria, we employed two methods. 1) OP50 bacteria in LB solution were killed by heating them for 1 h at 80 °C in a water bath. Death was confirmed by streaking this

solution on a LB plate, subsequently placing the plate for 2 days at 37 °C and verifying that no colony had formed at the end of this period. The solution of dead bacteria was then used to seed plates used for the lifespan. 2) NGM plates were seeded with live OP50 as usual and after 24 h, we covered the layer of bacteria with 100 μL of 10 mM kanamycin solution. Efficiency of bacteria lysis was confirmed by streaking bacteria from this layer onto a LB plate, subsequently placing this LB plate for 2 days at 37 °C and verifying no colony formation. NGM plates with the layer of kanamycin- killed bacteria were immediately used for lifespan experiment. Under both conditions, worms were transferred to new plates daily.

**Plate pretreatment with or without light**. NGM plates were seeded with OP50 and the next day placed in an incubator at 22 °C either under PD or NL conditions. Synchronized D1 adult worms, grown in permanent dark conditions at 22 °C, were transferred to these light- or dark-pretreated plates, and subsequently kept in NL conditions. Worms continued to be moved every day or every other day to similar light- or dark-pretreated plates until the end of their life and survival was recorded.

**Plate pretreatment with antioxidants**. N-Acetyl-cysteine (MilliporeSigma, Carlsbad, CA, USA) and vitamin C (MilliporeSigma, Carlsbad, CA, USA) were each resuspended in water and sterile filtered to obtain stock solutions at 1 M for both compounds. The appropriate volume of each stock solution (or the same volume of water for vehicle condition) was added into the freshly prepared liquid nematode growth media to achieve a final concentration of 5 mM (low), or 10 mM (high) for both compounds. Then, the liquid nematode growth media containing an antioxidant (or vehicle) was gently homogenized and immediately poured in small 60 × 15-mm Petri Dish plates (E&K Scientific, USA). After the media solidified, plates were kept at 4 °C for a maximum of 3 weeks until their use. Before starting lifespan analyses, worms were placed onto low-concentration NAC, high-concentration NAC, low-concentration VitC, or high-concentration VitC, or vehicle plates for at least two generations. For lifespan, worms were maintained on their respective antioxidant condition and synchronized using the egg-laying method. SD or bench light treatment started from D1 of adulthood. Worm survival scoring was performed every other day. At day 1 and day 5 of adulthood, worms were treated with 5-fluoro-2'deoxyuridine (FUdR) to prevent internal hatching as previously described[45].

**Temperature measurements**. Temperature was measured at noon on seeded NGM plates exposed to the four photoperiod conditions for 24 h. The precision thermometer (Acorn Temp TC Thermocouple Meter, Kent Scientific, Torrington, USA) equipped with a flexible Type-T temperature probe (±0.1 °C precision) was used to record the temperature on the surface of the agar of the NGM plate, and in the agar of the NGM plate. Six plates per condition were used and for each condition, measures were recorded three times per plate and averaged.

**Oxyblot**. Worms were hatched and grown on 100 mm NGM plates seeded with OP50. Day-1 adult worms were exposed either to NL or PD for up to 72 h. Worms at +36 h were rinsed and collected with M9 solution containing carbenicillin, transferred to Eppendorf tubes, and frozen in liquid nitrogen. Worms were crushed and ground up using a Teflon pestle, and the degree of protein oxidation was assessed using the OxyBlot protein oxidation detection kit following the manufacturer's instructions (Millipore S7150). A western blot of β-actin was run as loading control.

**Gene expression assays**. Gene expression analysis was performed as previously described[59]. Briefly, synchronized samples containing approximately 1000 worms per sample were all collected after 72 h of PD versus NL exposure from hatch. RNA was extracted using the freeze crack method and trizol/chloroform before being further purified using the RNeasy mini kit (Qiagen, Valencia, CA, USA). cDNA was synthesized from 1 μg of RNA using the QuantiTect kit (Qiagen). In all, $n =$ 3–6 biological repeats were analyzed in triplicates (technical repeats) using SYBR green, an Applied Biosystems QPCR instrument, and the standard curve method. Expression was normalized using the geometric mean of the three housekeeping genes: *cdc-42*, *pmp-3*, and Y45F10.D4[60]. The Roche Universal ProbeFinder online tool was used to design primers. Primers used to detect each transcript are detailed in Supplementary Table 3.

**Brood size**. The total brood size was measured by singly plating late L4 worms under PD or NL conditions. Each adult worm was then transferred to a new plate every 12 h and the previous plate was kept at 20 °C in the dark for another 48 h when the number of alive progeny, visible as L3–L4 larvae was scored. This procedure was repeated until no alive progenies were counted after 4 consecutive times (48-h period). The progenies of eight worms were counted per light condition and means were compared using Student's t test.

**Dauer assay**. *daf-2(e1370)* and *daf-2(e1370)xdaf-16(mu86)* young adults were transferred to seeded NG plates and allowed to lay eggs for 4 h at 15 °C. Adults were removed and plates were placed at 22 °C under permanent light or no-light conditions. After 72 h, worms were observed under a dissecting

microscope and scored as dauer or adults. For each condition, $n = 250$–464 worms were scored.

**Fluorescent microscopy**. Transgenic worms were anesthetized using 1 mM levamisole solution (MilliporeSigma, Carlsbad, CA, USA) and aligned immediately on a nonseeded NG plate. Photomicrographs were acquired from DIC or GFP filter using a Leica S6E dissecting microscope equipped with a Leica digital camera (Leica Microsystems, Buffalo Grove, IL, USA). Alternatively, worms were anesthetized using 1 mM levamisole solution and immediately mounted on an agarose pad preset on a microscope slide. Vaseline was used to seal the coverslip and worms were pictured under DIC or GFP fluorescence using a Zeiss Axio Observer Z1 equipped with an Axiocam (Carl Zeiss Microscopy, Thornwood, NY, USA). GFP fluorescence mean intensity was analyzed using Image J software. When comparing fluorescence between samples, only nonsaturated pictures using fixed times of exposure were taken.

**Mitochondrial integrity**. GFP fluorescent microscopy pictures of *myo-3*::GFP$^{MT}$ worms exposed during 7 days from D1 adult to PD or NL condition were used for evaluating mitochondrial integrity[46, 47]. $n = 10$ worms per condition were used and one representative picture of the body wall muscles was taken per worm. Mitochondrial integrity was evaluated by an experimenter, blind of the conditions, as follows: (a) normal = undamaged mitochondria with a pattern of neatly stacked arrays aligned with muscle fibers; (b) mitochondria with some defects, visible as thicker and often-fragmented arrays; and (c) severely damaged mitochondria with high fragmentation and almost complete absence of a consistent pattern of alignment with muscle fibers. Differences between PD and NL were then assessed using Fisher's exact test.

**Body size**. Worm body surface was calculated for D1 adult worms exposed to different light conditions from hatch. Micrographs were obtained from worms anesthetized and aligned on empty NGM plates as described above. Image J software was used to measure the body surface of individual worms. $n = 35$ worms per condition were used.

**Statistical analyses**. Data were analyzed using SPSS Statistics software 21.0 (IBM, Armonk, USA). Results are presented as mean ± standard error of the mean (SEM). For the data showing normal distribution under the Kolmogorov–Smirnov test, groups were compared using the two-tailed Student's t test. Non-normally distributed data were analyzed using the Mann-–Whitney test. Correlative statistics were performed using the Pearson product moment correlation coefficient. Lifespan curves were plotted using the Kaplan–Meier estimate and analyzed for significance between groups with the log-rank test. Levels of significance are $*P < 0.05$; $**P < 0.01$; $***P < 0.001$; $****P < 0.0001$; NS, not significant ($P > 0.05$). We also indicated the exact $P$ value when tendencies but not significance were found: $0.05 < P < 0.10$.

**Data availability**. The data that support the findings of this study are available from the corresponding author upon reasonable request.

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

## Acknowledgements
We thank Joanne Chory and Yogev Burko for access to different light incubators, Benjamin Cole for assistance with irradiance measures, Thuy N'Guyen for conception and assembly of LED devices, and Maryam Ahmadian for manuscript editing. C.D.M.F. was supported by the Howard Hughes Medical Institute and by a Glenn Foundation for Aging Research fellowship. L.R.L was supported by grants from the NIH (R00 AG042494, R01 AG051810), by a Junior Faculty Grant from the American Federation for Aging Research and by a Glenn Award from the Glenn Foundation for Medical Research. A.D. was supported by grants from the NIH (AG042679, AG024365, and AG027463), as well as by the Howard Hughes Medical Institute and the Glenn Foundation for Aging Research. R.M.E. supported this work with the Glenn Foundation for Aging Research grant.

## Author contributions

C.D.M.F. performed all the experiments except Oxyblot performed by N.E.S. and L.R.L. B.H. helped C.D.M.F. with the lifespan studies in R.M.E laboratory. A.D. and C.D.M.F. wrote the manuscript, and L.R.L. helped with comments on the manuscript.

## Additional information

**Competing interests:** The authors declare no competing financial interests.

