## [Peer Review File · Nature Communications]

Reviewers' comments:

Reviewer #1 (Remarks to the Author):

Major claims:

De Magalhaes Filho et al. show that exposure to light in the visible spectrum dramatically shortens the lifespan of *C. elegans*. This is independent of known circadian control genes and appears to be a function of allover light energy. The authors show that exposure to visible light triggers a generalized stress response and massive oxidative damage. They claim that known long-lived mutants that are stress resistant are specifically protected from light-induced damage. Finally, survival data from wild isolates are shown supporting the idea that resistance to light-induced damage is an adaptive strategy in environments with high light exposure.

The work contains two major novel findings. First, visible light induces significant biological damage and stress. Second, future work with *C. elegans*, particularly in aging and stress biology, will have to include considerations regarding phototoxicity. The findings will certainly trigger changes to regular protocols regarding handling of nematodes in the lab, which is relevant to others in the field.

Major points

- Oxidative stress: The authors show that exposure to visible light triggers massive oxidative stress and an associated stress response. Together, these correlate with a shortened lifespan. To test if oxidative stress is indeed causative for the pathology and to add some mechanistic insight, the authors should perform the survival experiments in the presence of antioxidants.

- Protection from long-lived mutants from light toxicity: *daf-2* mutants show protection in SD conditions, but under harsher LD and PD conditions, there appears to be no biologically relevant difference (Fig. 5a). Likewise, *eat-2* mutants show protection in SD, but the replicate in the supplement contradicts this result. *isp-1* mutants show only slight protection in SD. Given the current data set I wonder if the interpretation here should be that even long-lived mutants do not show significant protection from visible light-induced toxicity. The authors set out to do these experiments to understand the role of ROS in the longlived mutants. However, they do not measure the levels of ROS or the biological consequences of ROS, for example in an oxyblot.

- Protection of wild isolates: The data in Figure 5d is unconvincing. First, the lifespan effect is small and could not be reproduced in all cases. Second, proper controls are missing: the experiment should be performed under NL, SD, LD, and PD conditions. WT isolates might well differ in lifespan under normal culture conditions. To further support the correlation, the authors might consider also testing isolates from areas with shorter days. However, even if the correlation holds, the data is not strong enough to suggest a selective pressure and subsequent adaptation.

- UV radiation in visible light: Although I agree that UV-induced mutations will only play a minor role in the observed pathology, the experiment in Figure 3 does not properly address this point. The authors reason that UV damage, if responsible for the short lifespan, would be inherited. As survival is unaffected in the offspring of light-exposed worms, they conclude that UV does not contribute to the toxicity. Here, they do not consider the effect of somatic mutations as a driver of toxicity: light induced somatic DNA damage might cause the toxicity without affecting the germline, leaving the F1 generation, at a population level, intact. Furthermore, to test the mutagenic potential of irradiation, a lifespan experiment in the F1 generation is not informative: recessive mutations will only be homozygous in the F2.

- Biological replicates: it appears from the manuscript that many experiments were performed only once. Given the variability of *C. elegans* lifespan results even within one lab (see NL controls in Fig. 3 d vs e), survival experiments should be done in at least 3 biological repeats. In Fig. 5b, the second replicate (data in supplement) is in direct disagreement with the key point of the figure: *eat-2* mutants are not protected from SD light. Furthermore, the longevity of the wild isolates

shown in Fig. 5d could not be clearly replicated. This might be due to very low animal numbers in these experiments, which therefore should be repeated.

Minor points

- Details on a number of experimental procedures are missing: how were worms scored under the NL condition (how much light were they exposed to during scoring)?

- Nomenclature and labeling: What are the strains used in Fig. 1d and g? As JU361 is also used in the figure, it would be good to point out if it is N2. What are the black boxes in Fig. 1e? Nomenclature of gene names in Fig. 1f is not according to standards in the field. It is not clear how the experiment was done in Fig. 4a - what was the age of the animals? Labels for NL and PD are missing in Fig. 4a. ANOVA will be the more appropriate test for statistical significance in experiments with more than 2 conditions such as in Fig. 4 g and i.

- For the final paper, the lifespan analyses should be completed.

- Comments on text:

Page 5 states that JU431 was isolated "very close to England". Where was it isolated?

On page 10 the data from Figure 2 are interpreted as suggesting that photooxidation is a major driver of light-induced toxicity. Photooxidation is one of many potential mechanisms here. The survival data of Figure 2 per se do not suggest a role of photooxidation.

Legend of Figure 2: the title states that lifespan is decreased through a cell nonautonomous effect. There are no data to address cell autonomous or non-autonomous effects.

Taken together, the finding that visible light is toxic to nematodes is important for the work in any *C. elegans* lab. However, this finding in itself does not warrant publication in *Nature Communications*.

At the same time, the paper provides very interesting potential insights into the biology of stress and ageing. The finding that visible light triggers massive oxidative stress is remarkable and I think the paper is a good fit for the journal if this point could be further explored experimentally. Are ROS truly elevated upon exposure to visible light and can antioxidants protect from light-induced toxicity?

Reviewer #2 (Remarks to the Author):

In this manuscript, the authors show how light has a detrimental on *C. elegans* lifespan. They find that increasing time of white light exposure shortens nematode lifespan and that this is independent of genes involved in circadian rhythm. The authors also analyze different light sources for their potential to shorten nematode lifespan, and find that laboratory bench is sufficient to reduce the overall lifespan of *C. elegans*. Interestingly, light-treated nematodes exhibit increased stress responses in the endoplasmic reticulum and mitochondria and display an accumulation of oxidized proteins. The authors then go on to examine the impact of light on long-lived mutants (*daf-2* and *eat-2*) and on wild isolates from regions of the earth with a higher annual light intensity.

This study is very interesting and novel, as the effects of light on *C. elegans* lifespan and stress responses have never been examined before. As blue light can induce the stress response in cultured cells and induce the generation of ROS (Mamalis, 2015; Lavi, 2010), the responses observed in *C. elegans* may have important relevance in other species. Furthermore, this study is of great importance to the large *C. elegans* community (in the fields of development and aging). Indeed, this parameter could explain lifespan differences between labs. The impact of light on organismal physiological function is a very interesting field that should appeal to the large

audience of Nature Communications. The experiments presented in this manuscript are well-designed and compelling and the manuscript is well written.

It would be helpful to address the following points to help strengthen this already strong manuscript:

Major points:

1) Could the authors provide additional mechanistic insight in light-induced lifespan shortening? For example, it would be interesting to determine whether antioxidants restore lifespan under conditions of light exposure. Alternatively, the authors could use mutants in the redox pathway or in the UPR pathways and analyze their potential to resist (or be more sensitive) to light exposure.

2) The reduced impact of light on the lifespan of wild isolates from highly sun-exposed areas is potentially interesting. However, it is not entirely compelling at this current stage. Wild isolates might have different lifespan due to other parameters. The authors should assess the lifespan of N2 vs other isolates in NL versus PD (or SD or LD) conditions and compare the differences. It could also be helpful to test isolates from more dark places in Earth. Alternatively, this panel, while interesting, could also be removed, as it is not essential for the main message of the paper.

3) Figure 4, RT-qPCR, statistical tests: it would probably be more appropriate to use statistical tests that do not assume that the data are normally distributed (for example two-tailed Mann-Whitney test) instead of a Student's t-test. The number of samples seems too low to conclude that the data are normally distributed.

Minor points:

1) Figure 5: The authors conclude that long-lived mutants are more resistant to light stress than wild-type. However, this increased resistance is not very strong. Could the authors perform ANOVA to test if the differences in lifespan between WT NL vs. SD and long-lived mutants NL vs. SD are significantly reduced?

2) Figure 2c: the lifespan curves look different than other curves: were they done in a different way? Could they be presented in the same way?

3) It would be very informative to include a table for Figure 3a and b (the values for ambient room light are not clearly identifiable in these). This would help the *C. elegans* community identify the light stress in their lab environment and compare it to the authors' settings.

Reviewer #3 (Remarks to the Author):

In this manuscript, the authors reported an interesting finding that visible light shortens worm lifespan. It is the intensity, rather than wavelength of the light, that determines the lifespan-shortening effect. In addition, this effect is mediated by photo-oxidation, and is independent of photosensation mechanisms in worms. Long-lived mutant worms were found to be more resistant to light exposure. Interestingly, wild-type strains isolated from areas with longer day-time are likely to have longer lifespan than Bristol N2 worms. To my great surprise, the authors found that even scoring and passaging worms with ordinary lab light exposure could significantly shorten

lifespan. As far as I know, this is the first report showing that visible light has such a strong effect on *C. elegans* physiology. Overall, this is a very interesting study, and my enthusiasm is high. A few questions need to be addressed before publication:

- 1, *lite-1(ok530)* allele is very strange. It is a deletion but also a translocation, namely, a copy of *lite-1* gene is translocated elsewhere in the genome. This allele is probably not a null. Please test *xu7* and *ce314*.
2. For stress measurements, please show the effect of LD and SD treatments, as PD treatment is so strong and 24hrs light-on hardly occurs in the wild and in the lab.
3. As the authors suggest that photo-oxidation would be the major driver for visible light to decrease the lifespan of *C. elegans*, do anti-oxidants antagonize the life-shortening effect of visible light?
4. Please include more replicates for Figure 5d, as the data shown in the main figure is quite different from the data shown in the supple table.

Reviewer #4 (Remarks to the Author):

In this paper, the authors demonstrate that white light, at intensities similar to normal daylight, cause a significant shortening of lifespan and an increase in oxidative stress. They further show that this effect is greater using higher-energy, shorter wavelength light.

In general, this is an interesting finding, as to my knowledge a major effect of light on worm aging was unsuspected. I imagine (not being an aging person) that these results will impact how investigators studying aging conduct their experiments in the future. However, the effects of light at intensity levels comparable to a typical lab environment are much more modest, and effects on other aspects of worm biology, especially on young adult worms, are not addressed here. Thus, it is unclear how much impact these findings will have on worm experimentation outside the aging/stress field.

Other comments:

The authors find that four wild *C. elegans* strains isolated from tropical latitudes live slightly longer in high light conditions than the N2 lab strain. This finding seems a bit too anecdotal to make a strong argument for evolutionary selection. Only four strains were tested, all from more equatorial latitudes than N2. Moreover, N2 is a lab strain that has been effectively domesticated for many generations. To make a strong argument for natural selection of light resistance, the authors really should sample a greater number of strains, from a range of latitudes, and assess the strength of the correlation.

The authors also state that "*daf-2* worms were protected from light insult". But it seems like the *daf-2* mutants proportionally were comparably affected by light to N2, they just live longer under all conditions. I think the authors should reword their interpretation of this experiment to reflect the fact that the effects of light and *daf-2* on aging seem to be more or less independent.

Reviewers' comments:

Reviewer #1 (Remarks to the Author):

Major claims:

De Magalhaes Filho et al. show that exposure to light in the visible spectrum dramatically shortens the lifespan of *C. elegans*. This is independent of known circadian control genes and appears to be a function of allover light energy. The authors show that exposure to visible light triggers a generalized stress response and massive oxidative damage. They claim that known long-lived mutants that are stress resistant are specifically protected from light-induced damage. Finally, survival data from wild isolates are shown supporting the idea that resistance to light-induced damage is an adaptive strategy in environments with high light exposure. The work contains two major novel findings. First, visible light induces significant biological damage and stress. Second, future work with *C. elegans*, particularly in aging and stress biology, will have to include considerations regarding phototoxicity. The findings will certainly trigger changes to regular protocols regarding handling of nematodes in the lab, which is relevant to others in the field.

Major points

- Oxidative stress: The authors show that exposure to visible light triggers massive oxidative stress and an associated stress response. Together, these correlate with a shortened lifespan. To test if oxidative stress is indeed causative for the pathology and to add some mechanistic insight, the authors should perform the survival experiments in the presence of antioxidants.

→ We fully agree with the reviewer that experiments using antioxidants to rescue the shorter lifespan under light exposure will strengthen our demonstration that visible light creates oxidative stress in *C. elegans*. Therefore, we treated N2 worms from hatch with N-acetyl cysteine (NAC) or Vitamin C (VitC) at different concentrations previously shown to confer resistance to oxidative stress, and exposed these worms to visible white light from D1 adult. These results are now included in Figure 5 as new panels c and d. First, they confirm that the presence of antioxidants can rescue the short lifespan of N2 worms exposed to SD light (panel c). In accordance with this observation, the lifespan rescue with antioxidants was even more evident when exposing worms to a weaker white light intensity. Using the bench light condition, previously used in Figure 2 panel e, worms exposed to bench light while on plates supplemented with NAC at high or low concentrations, or Vitamin C at high or low concentrations all demonstrate a significantly longer lifespan than worms not receiving antioxidants (+30 to +105% increase in mean lifespan, Figure 5 d). Strikingly, when comparing the lifespan of no light control worms to the lifespan of worms under bench light with antioxidants, we note that Vitamin C at low concentration almost completely rescues the lifespan of bench light worms and NAC at high concentrations completely rescues it.

- Protection from long-lived mutants from light toxicity: *daf-2* mutants show protection in SD conditions, but under harsher LD and PD conditions, there appears to be no biologically relevant difference (Fig. 5a). Likewise, *eat-2* mutants show protection in SD, but the replicate in the supplement contradicts this result. *isp-1* mutants show only slight protection in SD. Given the current data set I wonder if the interpretation here should be that even long-lived mutants do not show significant protection from visible light-induced

toxicity. The authors set out to do these experiments to understand the role of ROS in the longlived mutants. However, they do not measure the levels of ROS or the biological consequences of ROS, for example in an oxyblot.

→ *daf-2* worms live significantly longer than N2 worms not only under SD conditions but also under LD conditions ($P < 0.0001$, Figure 5a, Supp Fig 6a and Supp Table 2) and PD conditions ($P < 0.0001$, Figure 5a Supp Fig 6a and Supp Table 2). We repeated these lifespan experiments several times and all repeats were consistent in showing an increase of lifespan in *daf-2* mutant worms. The repeats show that the increase under PD and LD conditions are even higher than previously observed. We therefore replaced the Figure 5a with our new repeat curve and report all data in the Supplementary Figure 6a and Supplementary Table 2.

Regarding the *isp-1* mutants, we first realized that we indicated in the initial manuscript that “under LD or PD conditions, the lifespan of *isp-1* mutants and N2 were indistinguishable (Figure 5c)”. This was a mistake since the former Figure 5c and the Supplementary Table 2 were actually showing that even under LD and PD, *isp-1* mutants were long-lived compared to N2 worms. We have now repeated these lifespan experiments several times and still observed the same results: increased lifespan of *isp-1* mutants under the different light conditions. These results can be found in the new Figure 5b, and we report all data in the Supplementary Figure 6b and Supplementary Table 2.

Despite multiple repeats, we have been unable to repeat our initial experiments performed in 2010 on the *eat-2* mutant animals and have removed this data from the paper and text.

- Protection of wild isolates: The data in Figure 5d is unconvincing. First, the lifespan effect is small and could not be reproduced in all cases. Second, proper controls are missing: the experiment should be performed under NL, SD, LD, and PD conditions. WT isolates might well differ in lifespan under normal culture conditions. To further support the correlation, the authors might consider also testing isolates from areas with shorter days. However, even if the correlation holds, the data is not strong enough to suggest a selective pressure and subsequent adaptation.

→ We agree that more work should be done to strengthen our hypothesis that wild isolates might have evolved to resist the oxidative stress caused by light depending on the area where they reside. We have now removed this section for further investigation.

- UV radiation in visible light: Although I agree that UV-induced mutations will only play a minor role in the observed pathology, the experiment in Figure 3 does not properly address this point. The authors reason that UV damage, if responsible for the short lifespan, would be inherited. As survival is unaffected in the offspring of light-exposed worms, they conclude that UV does not contribute to the toxicity. Here, they do not consider the effect of somatic mutations as a driver of toxicity: light induced somatic DNA damage might cause the toxicity without affecting the germline, leaving the F1 generation, at a population level, intact. Furthermore, to test the mutagenic potential of irradiation, a lifespan experiment in the F1 generation is not informative: recessive mutations will only be homozygous in the F2.

→ Somatic mutations caused by UV could be a driver of toxicity. Should it be the case, we have to agree that these somatic mutations are heterozygous since there is no division of somatic cells in the worm. Therefore the F1 generation, which also harbors heterozygous (versus homozygous) mutations should have its lifespan reduced. But as presented in Figure 3c, this is not the case. That is why we maintain that our initial hypothesis of measuring the lifespan of the F1 progeny to exclude a potential involvement of UV light in the short lifespan of worms exposed to white light is still valid. This is also supported by the fact that UV is well known to cause mutations to the germ line and is actually a method for mutagenesis (Grenwald and Horvitz, *Genetics*. 1980 Sep;96(1):147-64; and Kutscher LM, Shaham S. Forward and reverse mutagenesis in *C. elegans*. *WormBook*. 2014:1-26). Finally, as the LED experiments presented in Figure 3d, e, f, and g prove that exposing worms to different ranges of visible light are sufficient to decrease lifespan of worms, it complements the results of the Figure 3c and collectively allow us to exclude UV as a driver of the white light toxicity.

- Biological replicates: it appears from the manuscript that many experiments were performed only once. Given the variability of *C. elegans* lifespan results even within one lab (see NL controls in Fig. 3 d vs e), survival experiments should be done in at least 3 biological repeats. In Fig. 5b, the second replicate (data in supplement) is in direct disagreement with the key point of the figure: *eat-2* mutants are not protected from SD light. Furthermore, the longevity of the wild isolates shown in Fig. 5d could not be clearly replicated. This might be due to very low animal numbers in these experiments, which therefore should be repeated.

→ We agree and have included multiple repeats of almost all experiments and few repeated twice when results were extremely consistent.

Minor points

- Details on a number of experimental procedures are missing: how were worms scored under the NL condition (how much light were they exposed to during scoring)?

→ We have now extended the material and methods section regarding lifespan analysis, including how worms under the NL were scored.

- Nomenclature and labeling: What are the strains used in Fig. 1d and g? As JU361 is also used in the figure, it would be good to point out if it is N2. What are the black boxes in Fig. 1e? Nomenclature of gene names in Fig. 1f is not according to standards in the field. It is not clear how the experiment was done in Fig. 4a - what was the age of the animals? Labels for NL and PD are missing in Fig. 4a. ANOVA will be the more appropriate test for statistical significance in experiments with more than 2 conditions such as in Fig. 4 g and i.

→ We agree and made the respected changes and outline our use of additional statistical methods. Briefly, the strain used in Fig1d and g are N2 worms. It is now included in the legends of that figure. The black rectangles indicate known areas of *sod-3::GFP* expression where fluorescence was quantified, it is also now detailed in the legend of that figure. For the nomenclature of Fig1f, it has now been replaced with the proper nomenclature. For the experiment of Fig 4a, we have now detailed how the experiment was done in the material and methods, within the gene expression analysis paragraph. Also the labels for NL and PD have been added to Fig 4a, we thank the reviewer for noticing that. The use of ANOVA versus Student's test in case of groups of more than 2 conditions is subject to an intense debate among

biostatisticians depending whether you include the notion of control group. In shorts, if you want to compare three groups A, B, and C, for differences among each other, indeed ANOVA is the test of choice. However if you aim at finding whether a treatment or mutation creates a difference in your variable compared to a non-treated or wild-type group, a lot of statisticians will argue that the test of choice is a Student's t-test for each treated group or mutant versus the control. We do think that it is the method to apply because we want to compare, in the case of Figure 4g for example, whether light treatment creates a difference versus the NL condition, and whether the tunicamycin treatment creates a difference compared to the NL condition, that is why we have not performed ANOVA.

- For the final paper, the lifespan analyses should be completed.

→ In very rare instances, the last 5-10% surviving worms from a lifespan experiment condition could not be followed until their natural death because of censoring due to mold or technical problems. However in these experiments, the unfinished lifespans were all pertaining to the NL control condition and hence, the conclusion drawn from each of these experiments is totally unaffected by the few censored worms from the NL control.

- Comments on text: Page 5 states that JU431 was isolated "very close to England". Where was it isolated?

→ By JU431, we believe the reviewer refers to the JU361 wild isolate mentioned on page 5. This isolate originates from Franconville, a city just a few kilometers north of Paris. Latitude and longitude for this city are respectively 48° 59' 10" North and 02° 13' 47" East. Therefore, on the world scale, Franconville can be referred as being very close to Bristol (England) where N2 worms were isolated, since latitude and longitude of Bristol are respectively 51° 27' 16" North and 02° 35' 16" West.

On page 10 the data from Figure 2 are interpreted as suggesting that photooxidation is a major driver of light-induced toxicity. Photooxidation is one of many potential mechanisms here. The survival data of Figure 2 per se do not suggest a role of photooxidation. Legend of Figure 2: the title states that lifespan is decreased through a cell nonautonomous effect. There are no data to address cell autonomous or non-autonomous effects.

→ We do "suggest" and not conclude after the Figure 2, that photooxidation is likely to be the major driver of the light-induced toxicity, and we provide additional experiments and their results in the following sections. At the end of figure 2 it is the most likely suggestion because the oxidative action of light is a well described phenomenon and by the end of figure 2 we have eliminated several other obvious other reasons, including a signaling pathway that would originate from light sensing by photoreceptors.

We removed the "cell autonomous" phrasing in the legend of Figure 2.

Taken together, the finding that visible light is toxic to nematodes is important for the work in any C. elegans lab. However, this finding in itself does not warrant publication in Nature Communications. At the same time, the paper provides very interesting potential insights into the biology of stress and ageing. The finding that visible light triggers massive oxidative stress is remarkable and I think the paper is a good fit for the journal if this point could be further explored experimentally. Are ROS truly elevated upon exposure to visible light and can antioxidants protect from light-induced toxicity?

→ Indeed ROS are truly elevated upon exposure to visible light, see Figure 3, and as you suggest, antioxidants do protect from light induced toxicity, see the new panels in Figure 5: panels c and d.

Reviewer #2 (Remarks to the Author):

In this manuscript, the authors show how light has a detrimental on *C. elegans* lifespan. They find that increasing time of white light exposure shortens nematode lifespan and that this is independent of genes involved in circadian rhythm. The authors also analyze different light sources for their potential to shorten nematode lifespan, and find that laboratory bench is sufficient to reduce the overall lifespan of *C. elegans*. Interestingly, light-treated nematodes exhibit increased stress responses in the endoplasmic reticulum and mitochondria and display an accumulation of oxidized proteins. The authors then go on to examine the impact of light on long-lived mutants (*daf-2* and *eat-2*) and on wild isolates from regions of the earth with a higher annual light intensity.

This study is very interesting and novel, as the effects of light on *C. elegans* lifespan and stress responses have never been examined before. As blue light can induce the stress response in cultured cells and induce the generation of ROS (Mamalis, 2015; Lavi, 2010), the responses observed in *C. elegans* may have important relevance in other species. Furthermore, this study is of great importance to the large *C. elegans* community (in the fields of development and aging). Indeed, this parameter could explain lifespan differences between labs. The impact of light on organismal physiological function is a very interesting field that should appeal to the large audience of Nature Communications. The experiments presented in this manuscript are well-designed and compelling and the manuscript is well written.

It would be helpful to address the following points to help strengthen this already strong manuscript:

Major points:

1) Could the authors provide additional mechanistic insight in light-induced lifespan shortening? For example, it would be interesting to determine whether antioxidants restore lifespan under conditions of light exposure. Alternatively, the authors could use mutants in the redox pathway or in the UPR pathways and analyze their potential to resist (or be more sensitive) to light exposure.

→ We fully agree with the reviewer that experiments using antioxidants to rescue the shorter lifespan under light exposure will strengthen our demonstration that visible light creates oxidative stress in *C. elegans*. Therefore, we treated N2 worms from hatch with N-acetyl cysteine (NAC) or Vitamin C (VitC) at different concentrations previously shown to confer resistance to oxidative stress, and exposed these worms to visible white light from D1 adult. These results are now included in Figure 5 as new panels c and d. First, they confirm that the presence of antioxidants can rescue the short lifespan of N2 worms exposed to SD light (panel c). In accordance with this observation, the lifespan rescue with antioxidants was even more evident when exposing worms to a weaker white light intensity. Using the bench light condition, previously used in Figure 2 panel e, worms exposed to bench light while on plates supplemented with NAC at high or low concentrations, or Vitamin C at high or low concentrations all demonstrate a significantly longer lifespan than worms not receiving antioxidants (+30 to +105% increase in mean lifespan, Figure 5 d). Strikingly, when comparing the lifespan of no light control worms to the lifespan of worms under bench light with antioxidants, we note that Vitamin C at low concentration almost completely rescues the lifespan of bench light worms and NAC at high concentrations completely rescues it.

2) The reduced impact of light on the lifespan of wild isolates from highly sun-exposed areas is potentially interesting. However, it is not entirely compelling at this current stage. Wild isolates might have different lifespan due to other parameters. The authors should assess the lifespan of N2 vs other isolates in NL versus PD (or SD or LD) conditions and compare the differences. It could also be helpful to test isolates from more dark places in Earth. Alternatively, this panel, while interesting, could also be removed, as it is not essential for the main message of the paper.

→ We agree that more work should be done to strengthen our hypothesis that wild isolates might have evolved to resist the oxidative stress cause by light depending on the area where they reside. We have now removed this section and will continue to explore.

3) Figure 4, RT-qPCR, statistical tests: it would probably be more appropriate to use statistical tests that do not assume that the data are normally distributed (for example two-tailed Mann-Whitney test) instead of a Student's t-test. The number of samples seems too low to conclude that the data are normally distributed.

→ As indicated in the statistical analyses section of the manuscript, we always perform a test of normality of the data, which is the Kolmogorov-Smirnov test, before choosing to run the Student's t-test, because otherwise, *ie* if data are not normally distributed, we use the Mann-Whitney test as you correctly propose. But in the case of our qPCR data, they are normally distributed.

Minor points:

1) Figure 5: The authors conclude that long-lived mutants are more resistant to light stress than wild-type. However, this increased resistance is not very strong. Could the authors perform ANOVA to test if the differences in lifespan between WT NL vs. SD and long-lived mutants NL vs. SD are significantly reduced?

→ *daf-2* worms live significantly longer than N2 worms not only under SD conditions but also under LD conditions ($P < 0.0001$, Figure 5a, Supp Fig 6a and Supp Table 2) and PD conditions ($P < 0.0001$, Figure 5a, Supp Fig 6a and Supp Table 2). We repeated these lifespan experiments several times and all repeats were consistent in showing an increase of lifespan in *daf-2* worms. The repeats show that the increase under PD and LD conditions are even higher than previously observed. We therefore replaced the Figure 5a with the new repeat curve and report all data in the supplementary Figure 6a and the supplementary table 2..

Regarding the *isp-1* mutants, we first realized that we indicated in the initial manuscript that “under LD or PD conditions, the lifespan of *isp-1* mutants and N2 were indistinguishable (Figure 5c)”. This was a mistake since the former Figure 5c and the Supplementary Table 2 were actually showing that even under LD and PD, *isp-1* mutants were long-lived compared to N2 worms. We have now repeated these lifespan experiments several times and still observed the same results: increased lifespan of *isp-1* mutants under the different light conditions. These results can be found in the new Figure 5b, and we report all data in the Supplementary Figure 6b and Supplementary Table 2.

Despite multiple repeats, we have been unable to repeat our initial experiments performed in 2010 on the *eat-2* mutant animals and have removed this data from the paper and text.

For the ANOVA proposed, we are not sure how we would be able to input our data to test the idea of differences in reduction of lifespan between two groups versus two other. We do not think that is feasible by any test. What would be the variable and what would be the samples? We can just describe to the best the percentage of reduction (or the absolute reduction) due to SD light in N2 versus in *daf2*, nothing more.

2) Figure 2c: the lifespan curves look different than other curves: were they done in a different way? Could they be presented in the same way?

→ No they were not done in a different way

3) It would be very informative to include a table for Figure 3a and b (the values for ambient room light are not clearly identifiable in these). This would help the *C. elegans* community identify the light stress in their lab environment and compare it to the authors' settings.

→ That is a very good idea, we have now added this data as a new Supplementary Figure 4.

Reviewer #3 (Remarks to the Author):

In this manuscript, the authors reported an interesting finding that visible light shortens worm lifespan. It is the intensity, rather than wavelength of the light, that determines the lifespan-shortening effect. In addition, this effect is mediated by photo-oxidation, and is independent of photosensation mechanisms in worms. Long-lived mutant worms were found to be more resistant to light exposure. Interestingly, wild-type strains isolated from areas with longer day-time are likely to have longer lifespan than Bristol N2 worms. To my great surprise, the authors found that even scoring and passaging worms with ordinary lab light exposure could significantly shorten lifespan. As far as I know, this is the first report showing that visible light has such a strong effect on *C. elegans* physiology. Overall, this is a very interesting study, and my enthusiasm is high. A few questions need to be addressed before publication:

1, *lite-1(ok530)* allele is very strange. It is a deletion but also a translocation, namely, a copy of *lite-1* gene is translocated elsewhere in the genome. This allele is probably not a null. Please test *xu7* and *ce314*.

→ In addition to the *lite-1(ok530)*, we did test whether *lite-1(xu7)* and *lite-1(ce314)* mutant worms were protected from the light toxicity. However, as shown in the new Supplementary Figure 3, lifespan of these worms was also strongly reduced upon light exposure indicating that the decrease in lifespan due to light exposure is not mediated through signaling via LITE-1 receptor.

2. For stress measurements, please show the effect of LD and SD treatments, as PD treatment is so strong and 24hrs light-on hardly occurs in the wild and in the lab.

→ We agree with the reviewer that the longer the photoperiod, the stronger the stress, and therefore the shorter the lifespan. That is why we chose to use a milder stress than the PD condition for the rescue experiments shown in the new panels of Figure 5c and d. Remarkably, we show in these panels that using the NAC or the Vitamin C antioxidants under SD and bench light is actually rescuing the stress induced by this lower light exposure and therefore increasing the lifespan.

3. As the authors suggest that photo-oxidation would be the major driver for visible light to decrease the lifespan of *C. elegans*, do anti-oxidants antagonize the life-shortening effect of visible light?

→ We fully agree with the reviewer that experiments using antioxidants to rescue the shorter lifespan under light exposure will strengthen our demonstration that visible light creates oxidative stress in *C. elegans*. Therefore, we treated N2 worms from hatch with N-acetyl cysteine (NAC) or Vitamin C (VitC) at different concentrations previously shown to confer resistance to oxidative stress, and exposed these worms to visible white light from D1 adult. These results are now included in Figure 5 as new panels c and d. First, they confirm that the presence of antioxidants can rescue the short lifespan of N2 worms exposed to SD light (panel c). In accordance with this observation, the lifespan rescue with antioxidants was even more evident when exposing worms to a weaker white light intensity. Using the bench light condition, previously used in Figure 2 panel e, worms exposed to bench light while on plates supplemented with NAC at high or low concentrations, or Vitamin C at high or low concentrations all demonstrate a significantly longer lifespan than worms not receiving antioxidants (+30 to +105% increase in mean lifespan, Figure 5 d). Strikingly, when comparing the lifespan of no light control worms to the lifespan of worms under bench light with antioxidants, we note that Vitamin C at low concentration almost completely rescues the lifespan of bench light worms and NAC at high concentrations completely rescues it.

4. Please include more replicates for Figure 5d, as the data shown in the main figure is quite different from the data shown in the supple table.

→ We think that more work should be done to strengthen our hypothesis that wild isolates might have evolved to resist the oxidative stress cause by light depending on the area where they reside. We have now removed this section and will continue testing this hypothesis.

Reviewer #4 (Remarks to the Author):

In this paper, the authors demonstrate that white light, at intensities similar to normal daylight, cause a significant shortening of lifespan and an increase in oxidative stress. They further show that this effect is greater using higher-energy, shorter wavelength light.

In general, this is an interesting finding, as to my knowledge a major effect of light on worm aging was unsuspected. I imagine (not being an aging person) that these results will impact how investigators studying aging conduct their experiments in the future. However, the effects of light at intensity levels comparable to a typical lab environment are much more modest, and effects on other aspects of worm biology, especially on young adult worms, are not addressed here. Thus, it is unclear how much impact these finding will have on worm experimentation outside the aging/stress field.

→ We agree that the stronger the light stress, the shorter the lifespan. However we do show in Figure 2e that the effects of light at intensity levels comparable to a typical lab environment are still very noticeable. Worms under bench light live 26% shorter than worms kept in the dark ($P < 0.001$ for all replicates). This is a very strong effect, likely to have already many effects on signaling pathways impacted by oxidative stress signaling in young animals. We agree with the reviewer that the data presented in our manuscript bring a lot of exciting questions including the extent of the laboratory light effects on the worm biology in young

animals. We are sure this will be the scope of future papers and overall we do think that our findings presented in this manuscript are very relevant not only for worm experimentation in the aging/stress field, but more generally for any laboratory using *C. elegans*.

Other comments:

The authors find that four wild *C. elegans* strains isolated from tropical latitudes live slightly longer in high light conditions than the N2 lab strain. This finding seems a bit too anecdotal to make a strong argument for evolutionary selection. Only four strains were tested, all from more equatorial latitudes than N2. Moreover, N2 is a lab strain that has been effectively domesticated for many generations. To make a strong argument for natural selection of light resistance, the authors really should sample a greater number of strains, from a range of latitudes, and assess the strength of the correlation.

→ We agree that more work should be done to strengthen our hypothesis that wild isolates might have evolved to resist the oxidative stress caused by light depending on the area where they reside. We have now removed this section and will continue this line of experiments for further validation.

The authors also state that "daf-2 worms were protected from light insult". But it seems like the daf-2 mutants proportionally were comparably affected by light to N2, they just live longer under all conditions. I think the authors should reword their interpretation of this experiment to reflect the fact that the effects of light and daf-2 on aging seem to be more or less independent.

We understand the point of the reviewer with the idea that we might compare proportion of lifespan reduction between NL and light insult instead of the relative increase of lifespan in mutants exposed to light versus N2 worms exposed to light.

daf-2(e1370) worms are protected from the light insult because as shown in our manuscript they survive longer the different levels of light insult than N2 worms. The light resistance is therefore one of the many stress resistances that the *daf-2* mutation confers along with paraquat resistance (Kenyon C *et al*, Nature. 1993 Dec 2; 366(6454):461-4; and Honda *et al*, FASEB J. 1999 Aug;13(11):1385-93), heavy-metal stress (Barsyte D, *et al*, FASEB J. 2001 Mar; 15(3):627-34), heat resistance (Lithgow GJ *et al*, PNAS. 1995 Aug 1; 92(16):7540-4), pathogen resistance (Garsin DA *et al*, Science. 2003 Jun 20; 300(5627):1921) and Abeta(1-42) toxicity protection (Cohen *et al*, Science. 2006 Sep 15;313(5793):1604-10), among others. Therefore, we agree with the reviewer that the "*daf-2* mutants might be comparably affected by light to the N2", but this does not take away that they are still resistant to this stress and to the other ones mentioned above when compared to wild type worms.

Reviewers' Comments:

Reviewer #1:

Remarks to the Author:

De Magalhaes Filho et al. have addressed the most relevant points I had previously raised. The manuscript is now convincing as the causal role of ROS is established through the experiments using antioxidants. Minor comment: Direct ROS measurements and/or oxyblots for the effects of genotype, light exposure, and antioxidants (in combination) would have further strengthened the point regarding a causal role of ROS.

Indeed, the comparative studies with the wild isolate strains was difficult to interpret and leaving it out makes for a stronger paper with a clear message. This paper will be important for all researchers handling *C. elegans*. We as a field will need to come to standard handling protocols regarding visible light exposure. Furthermore the paper gives novel insight into stress resistance and aging.

The manuscript has my support for publication.

Reviewer #2:

Remarks to the Author:

The authors have addressed the reviewers' comments in a compelling manner and the added experiments and analyses have significantly enhanced this interesting manuscript.

Reviewer #3:

Remarks to the Author:

The authors have fully addressed my comments. I am very happy to support its publication. It is a very interesting piece of work. Congratulations!

One minor point: in the Introduction, please at least include ASH neuron to the list of photosensory neurons. Four neurons: ASJ, ASK, AWB and ASH together are required for light-evoked head avoidance behavior (there are probably additional photosensory neurons).